# Association between an inflammatory biomarker score and future dementia diagnosis in the population-based UK Biobank cohort of 500,000 people

Krisztina Mekli[1]*, Artitaya Lophatananon[2], Asri Maharani[3], James Y. Nazroo[1], Kenneth R. Muir[2]

1 Cathie Marsh Institute and Sociology, The University of Manchester, Manchester, United Kingdom,
2 Division of Population Health, Health Services Research and Primary Care, The University of Manchester, Manchester, United Kingdom, 3 Department of Nursing, Manchester Metropolitan University, Manchester, United Kingdom

* krisztina.mekli@manchester.ac.uk

## Abstract

This study was designed to investigate the relationship between a systematic inflammatory biomarker measure, concurrent and later cognitive performance, and future dementia risk. The literature has reported the potential involvement of inflammation in cognitive performance as well as Alzheimer's Disease, but not consistently. We used a population-based cohort of 500,000 people in the UK and assessed the association between a composite inflammatory biomarker and cognitive performance measures across five domains measured concurrently and 4–13 years later, taking advantage of the large sample size. We also assessed the same biomarker's association with dementia diagnosis 3–11 years later in the initially dementia-free sample. We report small but significant associations between elevated biomarker levels and worsened cognitive performance at baseline for four cognitive tasks (OR = 1.204, p<0.001 for Prospective memory, β = -0.366, p<0.001 for Fluid intelligence, β = 8.819, p<0.001 for Reaction time, and β = -0.224, p<0.001 for Numeric memory), comparing the highest quartile of the biomarker to the lowest. We also found that for one measure (Pairs matching) higher biomarker levels were associated with fewer errors, *i.e.* better performance (β = -0.096, p<0.001). We also report that the 4th quartiles of the baseline biomarker levels were significantly associated with cognitive task scores assessed years later on the p< = 0.002 level, except for the Pair matching test, for which none of the quartiles remained a significant predictor. Finally, the highest biomarker quartile was significantly associated with increased dementia risk compared to the lowest quartile (HR = 1.349, p<0.001). A case-only analysis to assess disease subtype heterogeneity suggested probable differences in the association with the highest biomarker quartile between vascular dementia and Alzheimer disease subtypes (OR = 1.483, p = 0.055). Our results indicate that systemic inflammation may play a small but significant part in dementia pathophysiology, especially in vascular dementia.

**Data Availability Statement:** Data was drawn from the publicly available UKB study (https://www.ukbiobank.ac.uk/). Application number 5864.

**Funding:** This work was funded by the Advantage Foundation. The funders had no role in study design, data collection and analysis, decision to publish, or preparation of the manuscript.

**Competing interests:** The authors have declared that no competing interests exist.

## Introduction

The number of people worldwide afflicted with Alzheimer's disease (AD), or dementia of other types, is high—the AD number being estimated to be currently at least 30 million, and by 2150 predicted to exceed 152 million. Due to its significant healthcare consequences, intensive research has been carried out to identify the pathways behind this disease. Particular attention has been paid to the *APOE* gene, whose *ε4* allele is the main genetic risk factor for late-onset AD. Overall impairment in β-amyloid peptide (Aβ) clearance is probably a major contributor to disease development, and Aβ deposition in the form of senile plaques is more abundant in *APOE* ε4 carriers than in noncarriers [1]. Research suggests that amyloid accumulation may be necessary at the beginning of the AD cascade but not sufficient and there are downstream factors with a key role, such as neuroinflammation and tau accumulation in the pathological process [2].

Searching for other mechanisms highlighted the potential role of neuroinflammation in dementia pathology. Measuring proinflammatory and anti-inflammatory cytokines in the cerebrospinal fluids and plasma of AD and mild cognitive impairment patients in small studies with typically less than 100 cases gave controversial or inconclusive results [3, 4]. However, recent large-scale meta-analyses support the hypothesis of immune dysregulation in AD and mild cognitive impairment, reporting elevated peripheral levels of CRP, IL6 and IL1β in AD compared to controls [5, 6]. Dividing dementia according to aetiology, a moderate to large elevation of both blood IL6 and TNFα levels compared to healthy controls was associated with a vascular dementia diagnosis. Blood IL6 levels significantly differed between vascular dementia patients and AD patients, and higher IL6 levels were also associated with incident vascular dementia in a meta-analysis of 20 studies [7].

In the current study we investigated potential associations between systemic inflammation and dementia, using a composite systemic inflammatory biomarker score and evaluated its association with cognition in a large population-based study, the UK Biobank (UKB) (https://www.ukbiobank.ac.uk/). Previously a study using the UKB cohort of 500,000 people showed that baseline performances of four cognitive tests (Reaction time, Visual memory, Verbal-numerical reasoning, Prospective memory) were significantly predictive of incident dementia diagnosis three to eight years later, after adjustment for age, sex, and education with ORs of 1.27–3.28. These results were independent from constitutional (including genetic) risk factors as well as modifiable risk factors [8]. Therefore, we assessed the biomarker's association with cognitive performance measured at the same time and 4–13 years later and hypothesised that inflammatory biomarker levels are negatively associated with the individual's performance on these cognitive tasks.

We further hypothesised that given the association with the individual tests' association with later dementia, higher levels of inflammation would be associated with future dementia risk. The sample we used was dementia-free at baseline with future dementia diagnosis available from hospital records (International Disease Classification 10 and 9) and primary care records.

We intended this study as hypothesis generating and offered a potential mechanism to explain the findings.

## Materials and methods

### Study population

The UKB is a national cohort of 502,650 individuals, both males and females. Participants were recruited in 2006–2010, aged 40–69 years at the time and continue to be longitudinally

followed to capture subsequent health events. More details can be found at https://www.ukbiobank.ac.uk/. Participants consented to the UK biobank for their data/samples to be used for health-related research purposes. Ethics approved for the UK biobank was obtained from the North West- Haydock Research Ethics Committee (REC reference: 16/NW/0274). More details can be found at https://www.ukbiobank.ac.uk/learn-more-about-uk-biobank/about-us/ethics.

We categorised the participants according to their ethnic background using the UKB Ethnicity variable (UKB Field Code 21000, https://biobank.ndph.ox.ac.uk/showcase/field.cgi?id=21000) at baseline. We developed three categories: White (if individuals were British, Irish or other White background), South Asian (if individuals were Indian, Pakistani or Bangladeshi) and Black (if they were Caribbean, African or other Black background).

We assessed cardiovascular problems at baseline using the UKB Vascular/heart problem diagnosed by doctor variable (UKB Field Code 6150, https://biobank.ndph.ox.ac.uk/showcase/field.cgi?id=6150). Individuals were coded as having a cardiovascular problem if they had any of the following conditions diagnosed by a doctor: heart attack, angina, stroke or high blood pressure, irrespective of the number of problems reported. The cardiovascular problems variable was used in Instance 2, by updating the baseline variable by adding the same conditions reported after baseline, up to Instance 2.

For material deprivation we used the Townsend Deprivation Index at baseline as continuous variable (UKB Field Code: 22189, https://biobank.ndph.ox.ac.uk/showcase/field.cgi?id=22189).

## Biomarker data

We developed an inflammatory biomarker score following guidance from Morrison *et al.* [9]. We used White blood (leucocyte) cell count [10^9 cells/Litre] and C-Reactive Protein [mg/L] (UKB variables: Data-Field 30000 (ox.ac.uk) and Data-Field 30710 (ox.ac.uk)) measures at baseline. These biomarkers were analysed together using a standardised z-score. To compute the z-score, each biomarker was normalised using a z-transformation. Z-score was computed for each participant based on the subject's biomarker levels (x), study mean (μ) and study standard deviation (σ).

We computed the z-scores thus z-score = (x − μ)/σ. The combined z-score was the sum of the subject's individual WBC and CRP z-scores [9]. We only used the combined z-scores within 3SD of study mean to exclude participants with acute inflammation status (n = 6,466, 1.42%). Finally, we divided the sample into z-score quartiles, which were used in the association analyses.

We developed separate quartile variables for CRP and WBC using the same method.
We also used the original continuous composite score.

## Cognitive performance

There were five cognitive tests included in the UKB dataset at baseline. They were administered via a computerised touchscreen interface. Except for the Pairs matching and Reaction time tasks at baseline, they were not administered for the whole sample.

These tasks we repeated 4–13 years later at the Instance 2 imaging visit in a subsample of 18,000–28,000 participants.

*Prospective memory* (UKB variable: Data-Field 20018 (ox.ac.uk)).

Before any of the other cognitive tests were performed, the following text appeared on the computer screen: "At the end of the games we will show you four coloured shapes and ask you

to touch the Blue Square. However, to test your memory, we want you to actually touch the Orange Circle instead".

We rescored this variable as zero or one, depending on whether the participant completed the task successfully on first attempt or not.

*Fluid intelligence* (UKB variable: Data-Field 20016 (ox.ac.uk)).

This task consists of 13 logic/reasoning type questions to be answered within a 2-minute time limit. The variable scores the number of the correct answers. Participants who did not answer all of the questions within the allotted 2-minute limit are scored as zero for each unattempted question.

*Reaction time* (UKB variable: Data-Field 20023 (ox.ac.uk)).

The Snap game is designed to test reaction time (*i.e.* simple processing speed) by pressing a button as soon as two identical cards are seen on the touchscreen. This exercise involved 12 pairs of cards. The variable is the mean duration in milliseconds to the first press of snap-button summed over rounds in which both cards matched.

*Pairs matching* (UKB variable: Data-Field 399 (ox.ac.uk)).

In this task, participants were asked to memorise the position of 6 card pairs and then match them. As this variable shows the number of incorrect matches, the higher value means poorer cognitive performance.

*Numeric memory* (UKB variable: Data-Field 4282 (ox.ac.uk)).

The Numeric memory test was performed in the pilot phase of recruitment into the study and is only available for a subset of participants. In this test, participants were asked to recall numbers after a short pause. The test started with a 2-digit number and became 1-digit longer each time the participant remembered correctly (up to a maximum of 12 digits). The score is the maximum of digits remembered correctly.

If the participant abandoned the task, their results were excluded.

**Dementia case identification.** The International Disease Classification (ICD) 10 and 9 codes for dementia were obtained from the publication by Wilkinson *et al.* [10]. The ICD 10 has 212 data fields (follow-up data), and the ICD 9 has 46 data fields (follow-up data). Our analysis used data available up to 31$^{st}$ January 2020. Information on the date when the codes were recorded was available for each follow-up. For subjects with any of the dementia codes appearing more than once, the earliest diagnosis date was used.

*Primary care record linkage.* Data from Primary Care linkage was available in 45% of the UKB participants at the time of this analysis. There are two versions of medical Read codes available in the UKB: version 2 (v2) and version 3 (ctV3 or v3). Both versions provide a standard vocabulary for clinicians to record patient findings and procedures, in health and social care IT systems across primary and secondary care within the National Health Service (NHS) in the UK. First, we applied the dementia medical Read code version 2 listed in the article by Wilkinson *et al.* [10]. We further mapped Read code version 2 with version 3 using the mapping file. This mapping file was provided by the UKB. The mapping file allows the specific code to be mapped across different platforms. We then generated Structured Query Language to extract data from the UKB portal. The date when dementia was recorded was also extracted. This enabled us to define whether the case was an incident or a prevalent one. For individuals where dementia codes appeared more than once, the record with the earliest date was kept (first time of diagnosis).

*Criteria for case and control identification.* Participants were classified into one of the following categories: incident or prevalent dementia cases and controls. For incident cases, subjects had to fulfil both of the following criteria 1) dementia diagnosis occurred 3 years or more after baseline and 2) subjects with a dementia code from any sources. Prevalent cases had dementia diagnoses prior to entering the UKB study.

Subjects with no dementia code from any sources were coded as controls.

This work was carried out under UKB application number 5864.

### Genetics data

*APOE ε* genotypes were estimated using the UKB imputed genetics data using Plink version 1.9 [11]. Individuals with the *APOE ε2/APOE ε4* (protective/risk) genotypes were excluded from the analysis.

### Statistical analysis

We performed logistic regression for the Prospective memory task, with the z-transformed combined inflammatory biomarker score at baseline as the independent variable. Age when the task had been performed, sex (UKB variable: Data-Field 31 (ox.ac.uk)), *APOE* genotype (*ε4* carriers vs non-carriers), cardiovascular problems at baseline or up to Instance 2, ethnicity and Townsend Deprivation Index at baseline were used as the covariates. We also performed logistic regression using Alzheimer disease cases as a reference category in order to test whether higher systemic inflammation levels are associated with increased risk of having vascular dementia compared to Alzheimer's disease.

We estimated Odds Ratios (OR) in these analyses.

The same biomarker scores were used as the independent variable for the other four cognitive tasks and beta coefficients were estimated in linear regression analyses with adjustment for age at the time of the cognitive task performed at baseline, sex, *APOE* genotype, cardiovascular problems at baseline or up to Instance 2, ethnicity and Townsend Deprivation Index.

We further performed a Cox proportional hazard model to examine the multivariate association between biomarker measures at baseline and the risk of dementia with year as the time scale. Respondents with dementia at entry into the cohort were excluded from the analysis. The respondents were censored at the year of the last follow-up. As with the logistic regression analyses, we included biomarker quartiles, sex, *APOE ε4* genotype, cardiovascular problems at baseline, ethnicity, and Townsend Deprivation index at baseline as the covariates. We performed the Cox proportional hazard model separately for CRP and White Blood Cell count as well as *APOE ε4* non-carriers and carriers.

All statistical analyses were performed using Stata version 14.0 (https://www.stata.com/).

## Results

### Demographics

Of the 502,536 samples, in this study we included the incident dementia cases and controls, while excluding the prevalent cases and individuals with the *APOE ε2/APOE ε4* genotype. It yielded a sample of 433,556 individuals.

We had 430,463 controls and 3,093 dementia incident cases in the study sample. Of the 198,935 males 197,358 (99.21%) were in the controls and 1,577 (0.79%) were in the incident cases subsample. Of the 234,624 females 233,105 (99.35%) were in the control and 1,516 (0.65%) were in the incident cases subsample. The mean age was significantly different between the controls (65.40, SD = 8.10 and incident cases (69.55, SD = 6.22) subsamples as indicated by the t-test (t = -28.50, p<0.0001).

The sample characteristics can be seen in Table 1.

The age in the analysis was calculated as age at dementia diagnosis for the incident cases and age at last follow-up in 2017 for the controls. For the incident cases, the time gap between baseline and dementia diagnosis was 3–11 years (mean: 6.31, SD = 1.85).

**Table 1. Demographics of the sample.**

| | Biomarker quartile | | | |
| --- | --- | --- | --- | --- |
| | 1st | 2nd | 3rd | 4th |
| Dementia diagnosis | | | | |
| control n (%) | 107,608 (99.42) | 107,716 (99.33) | 107,570 (99.24) | 107,569 (99.16) |
| case n (%) | 629 (0.58) | 728 (0.67) | 822 (0.76) | 914 (0.84) |
| Age mean (SD) | 64.577 (7.985) | 65.438 (8.066) | 65.816 (8.103) | 65.847 (8.175) |
| Sex | | | | |
| male n (%) | 49,367 (45.61) | 50,693 (46.75) | 50,738 (46.81) | 48,137 (44.37) |
| female n (%) | 58,870 (54.39) | 57,751 (53.25) | 57,654 (53.19) | 60,346 (55.63) |
| Cardiovascular problems | | | | |
| control n (%) | 85,450 (79.19) | 79,353 (73.42) | 73,618 (68.14) | 65,742 (60.87) |
| case n (%) | 22,456 (20.81) | 28,724 (26.58) | 34,417 (31.86) | 42,267 (39.13) |
| Diabetes diagnosed by doctor | | | | |
| control n (%) | 104,964 (97.28) | 103,776 (96.06) | 102,004 (94.51) | 97,711 (91.53) |
| case n (%) | 2,935 (2.72) | 4,251 (3.94) | 5,931 (5.49) | 9,132 (8.47) |
| Ethnicity | | | | |
| White n (%) | 101,284 (96.27) | 102,830 (97.29) | 102,743 (97.22) | 102,159 (96.73) |
| South Asian (%) | 1,001 (0.95) | 1,494 (1.41) | 1,910 (1.81) | 2,399 (2.27) |
| Black (%) | 2,920 (2.78) | 1,374 (1.30) | 1,025 (0.97) | 1,057 (1.00) |
| Material deprivation | | | | |
| Townsend DI mean (SD) | -1.535 (2.985) | -1.545 (2.960) | -1.373 (3.049) | -0.861 (3.257) |
| *APOE ε4* carrier status | | | | |
| non-carrier n (%) | 76,441 (70.62) | 78,312 (72.21) | 80,075 (73.88) | 83,090 (76.59) |
| carrier n (%) | 31,796 (29.38) | 30,132 (27.79) | 28,317 (26.12) | 25,393 (23.41) |
| Baseline cognitive tests scores | | | | |
| Prospective memory | | | | |
| completed the task | 26,193 (78.22) | 28,063 (77.48) | 28,953 (76.76) | 28,575 (74.43) |
| did not complete the task | 7,293 (21.78) | 8,157 (22.52) | 8,768 (23.24) | 9,815 (25.57) |
| Fluid intelligence mean (SD) | 6.180 (2.153) | 6.087 (2.152) | 5.967 (2.148) | 5.770 (2.154) |
| Reaction time mean (SD) | 550.943 (113.796) | 555.969 (114.608) | 560.044 (116.366) | 567.305 (122.583) |
| Pairs matching mean (SD) | 4.155 (3.418) | 4.149 (3.383) | 4.151 (3.372) | 4.135 (3.373) |
| Numeric memory mean (SD) | 6.821 (1.302) | 6.746 (1.309) | 6.684 (1.328) | 6.531 (1.399) |

SD = Standard Deviation, Townsend DI = Townsend Deprivation Index at baseline

### Inflammatory biomarker score results

The mean of inflammatory biomarker score was -0.111 with the SD of 1.118 for the study sample, with the means being significantly different between the controls (0.112, SD = 1.117) and incident cases (0.049, SD = 0.165) (t-test t = -7.991, p<0.0001).

The ranges of the inflammatory biomarker scores for the subsample of 433,556 individuals who fell between +/- 3 Standard Deviation and had genetic data can be seen in Table 2.

Table 3 shows the association analysis results between five cognitive domains assessed at baseline and inflammatory biomarker scores levels measured at baseline. Age at baseline, sex, *APOE ε4* status, cardiovascular problems, ethnic background and material deprivation were included in these models.

Age, sex, cardiovascular problems, ethnic background and maternal deprivation were significant predictors in these models (p< = 0.001), whereas *APOE ε4* status was a significant predictor for Visual declarative memory (Beta = 0.028, p-value = 0.015).

**Table 2. Inflammatory biomarker quartiles in the UKB sample.**

| Quartile | Number of participants (%) | Mean | SD | Minimum of z-score | Maximum of z-score |
|---|---|---|---|---|---|
| 1st | 108,237 (24.96) | -1.298 | 0.311 | -3.669 | -0.888 |
| 2nd | 108,444 (25.02) | -0.598 | 0.166 | -0.887 | -0.307 |
| 3rd | 108,392 (25.00) | 0.033 | 0.212 | -0.307 | 0.437 |
| 4th | 108,483 (25.02) | 1.415 | 0.911 | 0.437 | 4.619 |

SD = Standard Deviation

For the Prospective memory task, we observed significant associations (p<0.001) between higher biomarker levels and worse performance in each quartile compared to the reference quartile. The effect sizes were small, the largest OR was 1.204 for the highest quartile.

Associations between the inflammatory marker score at baseline and cognitive performance were significant for the Fluid intelligence, Numeric memory and Processing speed tasks. The effect sizes for the second, third and fourth quartiles were -0.125, -0.229 and -0.366 (p<0.001) for the Fluid intelligence task and -0.062, -0.104 and -0.224 (p = <0.001) for the

**Table 3. Associations between inflammatory biomarker score quartiles and baseline cognitive tasks adjusted for age, sex, *APOE ε4* status, cardiovascular problems, ethnic background and Townsend Deprivation Index.**

| Predictors | OR | Coefficient | p-value | 95% CI lower | 95% CI upper |
|---|---|---|---|---|---|
| **Prospective memory (UKB Field Code 20018)** | | | | | |
| 1st quartile | Reference | | | | |
| 2nd quartile | 1.076 | | p<0.001 | 1.036 | 1.119 |
| 3rd quartile | 1.105 | | p<0.001 | 1.064 | 1.147 |
| 4th quartile | 1.204 | | p<0.001 | 1.160 | 1.250 |
| **Verbal and numerical reasoning (Fluid intelligence, UKB Field Code 20016)** | | | | | |
| 1st quartile | | Reference | | | |
| 2nd quartile | | -0.125 | p<0.001 | -0.156 | -0.093 |
| 3rd quartile | | -0.229 | p<0.001 | -0.261 | -0.197 |
| 4th quartile | | -0.366 | p<0.001 | -0.398 | -0.334 |
| **Processing speed (Reaction time, UKB Field Code 20023)** | | | | | |
| 1st quartile | | Reference | | | |
| 2nd quartile | | 2.533 | p<0.001 | 1.589 | 3.477 |
| 3rd quartile | | 4.227 | p<0.001 | 3.279 | 5.175 |
| 4th quartile | | 8.819 | p<0.001 | 7.860 | 9.777 |
| **Visual declarative memory (Pairs matching, UKB Field Code 399)** | | | | | |
| 1st quartile | | Reference | | | |
| 2nd quartile | | -0.032 | 0.027 | -0.061 | -0.004 |
| 3rd quartile | | -0.055 | p<0.001 | -0.084 | -0.027 |
| 4th quartile | | -0.096 | p<0.001 | -0.125 | -0.067 |
| **Working memory (Numeric memory, UKB Field Code 4282)** | | | | | |
| 1st quartile | | Reference | | | |
| 2nd quartile | | -0.062 | 0.001 | -0.098 | -0.026 |
| 3rd quartile | | -0.104 | p<0.001 | -0.140 | -0.069 |
| 4th quartile | | -0.224 | p<0.001 | -0.260 | -0.188 |

OR = Odds Ratio, CI = Confidence Interval

Numeric memory task. These effect sizes indicated increasingly worse cognitive performance for individuals with higher biomarker levels. For the Reaction time task higher biomarker levels were associated with significantly increasing reaction times in the quartiles, compared to the bottom quartile (beta values 2.533, 4.227, 8.819 with p-values<0.001).

Finally, for the Pairs matching task the beta values were -0.032, -0.055 and -0.096 (p = <0.027). This somewhat counterintuitive results indicated that higher inflammatory levels were associated with fewer errors, therefore better performance.

These results are presented with all the covariates in S1 Table.

Performing these analyses separately for the *APOE ε4* non-carrier and carrier subsamples we found that in the non-carrier subsample age, sex, cardiovascular problems, ethnicity and material deprivation were significant predictors on the p< = 0.001 level as before. This was the same in the carriers subsample with the exception of sex (p = 0.003) in the Prospective memory analysis, sex (p = 0.045) and cardiovascular problems (p = 0.227) in the Pairs matching analysis and cardiovascular problems (p = 0.003) in the Working memory analysis.

The effect sizes for the quartiles for the Prospective memory, Reaction time and Working memory tasks in the non-carriers subsample represented virtually no change compared to the whole study sample. On the other hand, in the carrier subsample the results remained significant on the p<0.001 level only for the 4th quartile with similar effect sizes to the whole sample results (OR = 1.179, β = 8.670, β = -0.162, respectively).

For the Fluid Intelligence task, we did not observe differences in the two subsamples' results compared to the whole sample analysis or to each other. The Pairs matching task results retained the pattern of higher biomarker quartiles' association with better performance in both the non-carriers and carriers subsample but with loss of significance. In both subsamples, only the highest quartile remained significant (β = -0.096, p<0.001, for the non-carriers and β = -0.097, p = 0.001, for the carriers subsample) which were similar effect sizes to the whole sample results. These results are presented in S2 Table.

We analysed the two biomarker scores separately applying the same model parameters to evaluate if the composite score performs better than the individual biomarkers. We found that individual scores tended to show slightly reduced effect sizes compared to the composite score (p<0.001). We also observed significance loss for the CRP only score in the 2nd quartile (Prospective memory, Reaction time and Working memory tasks) and for the WBC only score for the Working memory task. The exception was the Pair matching test results for which the CRP quartiles produced larger effect sizes than the composite score with the WBC scores retaining significance only in the 4th quartile. These results are presented in S3 Table.

In a separate analysis we controlled additionally for the time between time of baseline and sample processing, including a gap variable for the cognitive tasks analyses. This gap variable was developed as difference in years between sample taken and processed. The effect sizes did not change noticeably, although this variable was significant in all analyses (p<0.001) except the Working memory task (p = 0.899). (Data not shown).

We also used a continuous biomarker score for the cognitive tasks. Adjusting for sex, age, *APOE ε4* carrier status, cardiovascular problems, ethnicity and material deprivation as before we found the following results. For the Prospective memory task: OR = 1.062 (p<0.001), Fluid intelligence task β = -0.123 (p<0.001), Reaction time task β = 3.153 (p<0.001), Pairs matching task β = -0.031 (p<0.001) and Working memory task β = -0.072 (p<0.001).

To model the possible accelerating decline pattern of the cognitive function, we added the age squared variable in the regression models together with the previous ones. Age squared was significant in each analysis (p<0.001) without much decrease in the effect sizes of the quartiles or in their significance levels. The significant association between the two variables

and cognition showed that the trajectories of cognition in later life have curvilinear shapes. These results are presented with all the covariates in S4 Table.

We have included the interaction between age and sex in the analysis and the results show significant interactions between age and sex and Reaction time ($p<0.001$), Fluid intelligence ($p = 0.015$), Pairs matching ($p = 0.011$) and Working memory (0.021). These findings suggest that the changes of the four measures of cognitive functions among males as they get older differ from females.

We could not overcome the limitations of the cross-sectional design, but in a reduced sample in Instance 2 (Imaging visit after 2014), we could analyse the relationship between baseline inflammatory biomarker level quartiles and cognitive performances 4–13 years later. We used the same regression parameters, except for additionally adjusting for time elapsed between baseline and Instance 2 in years. The cardiovascular problems variable was updated with reported problems up to Instance 2. We found that the Pairs matching results were not significant for any quartiles anymore, and Reaction time results were not significant for quartiles 2 and 3. For the other tasks, the results remained significant on $p = 0.005$ level with similar effect sizes as in the concurrent association. The time between instances was also significant for each task ($p< = 0.005$), except for the Numeric memory task ($p = 0.471$).

Results are presented in S5 Table.

We performed pairwise correlations between the cognitive tasks. All correlations were significant ($p<0.001$) and ranged between -0.1296 (between Pairs matching and Working memory) to 0.3927 (between Fluid intelligence and Working memory). Pairs matching was also negatively correlated with Fluid intelligence (-0.1898), and positively correlated with Prospective memory (0.1565) and Reaction time (0.1376).

We also calculated pairwise correlations between the task scores measure at baseline and in Instance 2. The strongest correlation was between the Fluid intelligence scores (0.6371), as expected and the weakest was between the Pairs matching results (0.1843). Between the Prospective memory scores the correlation was 0.2535, whereas between the Reaction time scores it was 0.4903 and between the Working memory scores it was 0.5299. All correlations were significant on the $p = 0.001$ level.

Cox regression revealed that higher inflammatory biomarker scores were significantly associated with increased dementia risk in a model adjusted for sex, *APOE ε4* status, cardiovascular problems at baseline, ethnic background and Townsend Deprivation Index (Table 4).

**Table 4. Results of Cox regression of dementia diagnosis with sex, *APOE ε4* status, cardiovascular problems at baseline, ethnicity and Townsend Deprivation Index (TDI) included in the model.**

| Predictors | HR | p-value | 95% CI lower | 95% CI upper |
|---|---|---|---|---|
| 1st quartile | Reference | | | |
| 2nd quartile | 1.132 | 0.024 | 1.016 | 1.262 |
| 3rd quartile | 1.243 | p<0.001 | 1.118 | 1.381 |
| 4th quartile | 1.349 | p<0.001 | 1.215 | 1.498 |
| Sex | 1.144 | p<0.001 | 1.065 | 1.229 |
| *APOE* | 2.624 | p<0.001 | 2.443 | 2.818 |
| Cardiovascular problems | 1.890 | p<0.001 | 1.757 | 2.033 |
| Ethnicity | 1.027 | 0.667 | 0.909 | 1.161 |
| TDI | 1.033 | p<0.001 | 1.022 | 1.045 |

HR = Hazards Ratio, CI = Confidence Interval

**Table 5. Results of logistic regression for vascular dementia with sex, *APOE* ε4 status, cardiovascular problems at baseline, ethnicity and Townsend Deprivation Index (TDI) at baseline, using the Alzheimer disease category as reference.**

| Predictors | OR | p-value | 95% CI lower | 95% CI upper |
|---|---|---|---|---|
| 1st quartile | Reference | | | |
| 2nd quartile | 1.070 | 0.76 | 0.693 | 1.650 |
| 3rd quartile | 1.376 | 0.134 | 0.906 | 2.088 |
| 4th quartile | 1.483 | 0.055 | 0.991 | 2.220 |
| Sex | 1.637 | p<0.001 | 1.247 | 2.150 |
| Age | 1.014 | 0.391 | 0.983 | 1.046 |
| *APOE* | 0.609 | p<0.001 | 0.464 | 0.798 |
| Cardiovascular problems | 2.076 | p<0.001 | 1.569 | 2.747 |
| Ethnicity | 1.017 | 0.947 | 0.618 | 1.674 |
| TDI | 1.039 | 0.059 | 0.998 | 1.081 |

We fitted the same model separately to the *APOE ε4* non-carriers and carriers subsamples' data. Associations between inflammatory biomarker score quartiles and dementia risk remained significant only in the *APOε4* non-carrier subsample. Here, we observed slight increase in all HRs for all the biomarker quartiles compared to the whole sample (HR = 1.261, p = 0.004, HR = 1.330, P<0.001, HR = 1.570, p<0.001 for the 2nd, 3rd and 4th quartiles, respectively). All results can be seen in S6 Table.

Analysing the two biomarkers separately revealed that the biomarker dementia diagnosis association was driven by the leucocyte count because all quartiles remained significantly associated with the outcome compared to the reference quartile, whereas for CRP it was only true for highest quartile. Respondents in the 4th CRP quartile had a 1.148 higher risk of having dementia than those in the reference quartile (p = 0.006). Respondents in the 2nd, 3rd and 4th WBC quartiles had 1.199, 1.249, and 1.312 higher risk of having dementia than those in the first quartile (p< = 0.001).

Results are presented in S7 Table.

Finally, we analysed the sample according to dementia aetiology. We had 1,063 incident Alzheimer's disease cases and 462 incident vascular dementia cases in the sample. We performed logistic regression using the Alzheimer disease cases as reference category in order to evaluate if higher systemic inflammation levels are associated with increased risk of having vascular dementia compared to Alzheimer's disease. We adjusted the model for sex, *APOE ε4* status, cardiovascular problems at baseline, ethnicity and Townsend Deprivation Index (TDI) at baseline. In this analysis, only the 4th quartile showed marginally significant association with having vascular dementia (p = 0.055, OR = 1.483), whereas sex (OR = 1.637, p<0.001), cardiovascular problems (OR = 2.076, p<0.001) and APOE ε4 (OR = 0.609, p<0.001) possession were significantly associated with vascular dementia (Table 5).

## Discussion

In this study we observed small but significant associations between systemic inflammatory biomarker scores and future dementia diagnosis as well as worsened recent and subsequent cognitive performance in a large population-based sample. A previous study indicated the association of baseline cognitive tasks with dementia diagnosis 3–8 years later in the same cohort [8], therefore we assessed the association of five cognitive tasks with inflammatory biomarker levels, adjusting for age, sex and *APOE* genotype, cardiovascular problems, ethnicity, and material deprivation. We found that higher biomarker scores were significantly associated

with worse performance in each quartile for four of these cognitive tasks. While for the fifth task, Pairs matching, better performance was associated with higher inflammatory biomarker scores.

We observed that for most tasks the *APOE ε4* dosage was not significantly associated with the cognitive performance. Dividing the sample by *APOE ε4* carrier status, if it caused differences, it was in the carriers subsample as observed for the Prospective memory, Reaction time and Working memory tasks. It might indicate that the effect of inflammation is somewhat reduced in individuals carrying the *APOE ε4* allele. There is indication in the literature that *APOE ε4* has an effect on immune biomarker levels, including CRP [12]. However, this also might originate from sample size differences, having three times as many individuals in the non-carrier subsample. There was no or little change between the subsamples' results for the Fluid intelligence task, although the sample size difference was also present with three times as many non-carriers as carriers. *As APOE ε4* dosage was the least significant for this task of the five (p = 0.855), this might be the cognitive domain least influenced by *APOE*. For the Pairs matching task there was loss of significance in both subsamples. This was the only cognitive domain with the *APOE ε4* dosage being somewhat significant (p = 0.015) and the inflammatory quartiles reversely associated with the performance, meaning that the role of the former is more important. Taken together these results, they indicate that not all cognitive domains are affected by inflammation.

Being in the highest quartile for the inflammatory biomarker score, compared with the lowest quartile, was associated with an increased risk for dementia diagnosis by about 35% (HR 1.349, CI 1.215–1.498) in a fully adjusted model (including *APOE*). In this model the HR for *APOE* was significant and larger (HR = 2.624, CI 2.443–2.818) than that of the biomarker score's. Dividing the sample to *APOE ε4* non-carriers and carriers, yielded loss of significance for all the quartiles in the latter with a slight increase in the HR compared to the non-divided sample (HR = 1.57, CI 1.351–1.824). These results might indicate that inflammation contributes to dementia if *APOE ε4* is not present.

Analysing the biomarker scores separately indicated that it might be slightly more beneficial to use a composite score instead of individual ones. One biomarker can classify some cases correctly and misclassify others, therefore using a composite score can provide a more accurate inflammation status classification. For example, WBC and CRP although closely linked, proved to be independent predictors of mortality in the oldest old [13]. We observed some reduced effect sizes or significance loss of the individual scores compared to the composite, supporting the use of the composite score, and adding more components may yield an even more robust inflammatory score.

This analysis revealed that for Pairs matching WBC was only a significant predictor for the 4th quartile, whereas for CRP, there was some small increase in the effect sizes. For this domain CRP only appears to be the better predictor.

For the dementia diagnosis, the CRP-only score only remained a significant predictor for the highest quartile, whereas WBC appeared to be a more robust predictor with similar effect sizes to the composite score. Taken together these results, a composite score might provide a more accurate inflammation classification for most of the individual cognitive domains but not for all-cause dementia. For the latter the WBC-only score was sufficient.

Most of these tasks showed high correlations between the baseline and Instance 2 performance 4–13 years later. This might explain that why the baseline inflammation score was associated not just with concurrent but with subsequent performances measure. Pairs matching was an exception, with lower correlation between the instances and no significant association between the biomarker score at baseline and later cognitive performance.

We also found that the highest inflammatory biomarker score was marginally associated with vascular dementia using the AD subsample as reference category, although, having cardiovascular problems was associated with a higher risk.

Evidence in the literature supports the associations between inflammatory biomarkers and cognitive function, although the reports are somewhat inconsistent. For example, elevated serum CRP-levels were associated with cognitive decline in elderly women [14], and serum levels of CRP, Interleukin-6 and Interleukin-10 were negatively associated with a composite score of executive function and processing speed, but not with verbal episodic memory [15]. In contrast, Tampubolon using growth curve models reported that higher levels of CRP and fibrinogen had been associated with worse episodic memory, particularly among the older old [16]. The second phase of AD progression is the prodromal phase which is characterised by the onset of the earliest cognitive symptoms, typically deficits in episodic memory [17]. Therefore, the association between systemic inflammation and worsened episodic memory, as reported by Tampubolon, may be an indicator of an association with AD before it enters the third, last phase. We cannot confirm this association in this study as episodic memory was not assessed in UKB. As for dementia diagnosis, Song and Colleagues found significantly elevated CRP levels in patients with AD compared with healthy controls [18].

However, in a meta-analysis, compared with controls, only IL-1β was significantly elevated in Alzheimer's disease but not IL-6, TNF-α or CRP and the significance did not survive Bonferroni-correction [19]. The studies included in the meta-analysis had reported both elevated and decreased levels of biomarkers for the AD group. Small sample sizes (typically below 100) and heterogeneity in the studies, such as age range of the elderly, medical comorbidities, study design and methodological variances, probably contributed to the inconsistent results. Also, some authors suggest changing biomarker trajectories during disease progression. O'Bryant and colleagues reported significantly decreased mean CRP levels in the AD group in relation to controls, whereas within the AD group elevated CRP levels significantly predicted higher (poorer) dementia severity (Clinical Dementia Rating Scales) scores [20]. The Authors suggested that midlife elevations in CRP are a risk factor for the development of AD; however, this elevation appears to reduce and even fall below that of controls once the disease becomes clinically manifest.

Some authors propose that instead of a single inflammatory exposure, chronic low-grade inflammation (inflamm-aging) is the risk factor for age-associated diseases, such as dementia and frailty [21]. Ferrucci and Fabbri define inflamm-aging hypothesis as follows. 'Ageing is associated with immune dysregulations with high blood levels of pro-inflammatory mediators in the absence of evident triggers and, in parallel, reduced capacity to mount an effective inflammatory response to adequate immunogenic stimulations. The pro-inflammatory state is characterised by high circulating levels of pro-inflammatory markers, including IL-1, IL-6, C-reactive protein (CRP), tumour necrosis factor (TNF) among others. High levels of age-associated pro-inflammatory markers are detected in the majority of older individuals, even in the absence of risk factors and clinically active diseases' [22].

Longitudinal studies measuring inflammatory biomarkers in more than one time point are able to investigate this hypothesis. Tao and Colleagues provided evidence for the role of chronic low-grade inflammation in Alzheimer's Disease. They reported that the *APOE* ε4 allele coupled with chronic low-grade inflammation, defined as a CRP level of 8mg/L or higher, was associated with an increased risk of AD as well as an increased risk of earlier disease onset compared with *APOE* ε4 carriers without chronic inflammation [23]. Within the *APOE* ε4 group, increased levels of CRP were associated with increased risk of AD and dementia.

Metti and Colleagues constructed IL6 trajectories available from 3–6 time points and analysed trajectory properties in relation to cognition and brain measures. They found that

neither baseline levels nor slopes of IL-6 were related to cognitive impairment or hippocampal volumes, but IL-6 variability was positively associated with cognitive impairment and with a greater decrease of grey matter volume of the hippocampus [24]. Another study reported no significant correlation between change in CRP or IL-6 concentrations and cognitive decline [25].

Finally, Singh-Manoux and Colleagues examined the association between biomarker and cognitive measures using longitudinal data for both measures. They used the mean of two IL6 and CRP measures taken 5 years apart and cognitive tests administered at 3 clinical examinations over 10 years. In the cross-sectional analysis, reasoning was lower in participants with high compared to low IL-6. In the longitudinal analysis, 10-year decline in reasoning was greater among participants with high IL-6 than those with low IL-6. In addition, participants with high IL-6 had increased risk of decline, whereas CRP was not associated with decline in any test [26].

An important question is the relationship between cognitive decline and later dementia. Tampubolon and Colleagues showed that cognitive decline trajectories are strong predictors of dementia. They found that from all the factors investigated, by far the largest odds had been given by cognition trajectories; with the advantaged trajectory members having odds of one-fifth of the disadvantaged trajectory members to have dementia at the end of more than a decade of study [27].

Taken these results together, both single time point and longitudinal inflammatory biomarker measures provided evidence for associations between cognition and AD and inflammatory biomarker levels but not consistently. It is uncertain whether a single time-point inflammatory result is a flare or reflects long-term inflammatory exposure. Several studies indicated that CRP levels increase with age [23, 25]. This latter study also reported a positive association between increasing mean age and higher median CRP levels and a higher proportion of older participants having CRP levels of 3mg/L or higher across all *APOE* genotypes. This indicates that CRP-levels may not remain constantly low during ageing but increase, supporting the inflamm-aging hypothesis. In our study, comparing the mean inflammatory biomarker scores for the small subset of sample with repeated measures available up to 7 years later (n = 14,712) we found that the mean scores increased from baseline -0.284 (SD = 1.025) to -0.118 (SD = 0.972) with the difference being significant (t-test $p < 0.0001$). It raises the possibility that elevated inflammatory biomarker levels are maintained or even rise in ageing and, once these high levels reach a significant threshold it may trigger pathological mechanisms leading to cognitive decline or dementia.

In terms of mechanisms, it has been reported in the literature that CRP at clinically relevant concentrations (10–20 μg/mL) causes a disruption in the blood-brain barrier (BBB) in a Guinea pig cell coculture model [28]. The blood–brain barrier is responsible for maintaining the homeostasis of the central nervous system and protecting it from unwanted compounds. In human studies, associations have been shown between increased BBB permeability to small molecules, such as water (though not larger molecules such as albumin) and worsened cognitive states and risk of AD diagnosis [29]. However, measuring the cerebrospinal fluid/plasma albumin ratio an indicator of BBB permeability, Janelidze and Colleagues reported that the ratio was increased in all dementias compared with controls, but is not related to *APOE* genotype [30]. In contrast, another study reported significantly increased BBB permeability in the hippocampus in mild cognitive impairment compared to age-matched non-cognitively impaired controls. A significant 30% increase in the CSF/plasma albumin ratio further confirmed BBB breakdown in MCI individuals compared to age-matched NCI controls [31].

A possible mechanism to explain our results and the results reported in a literature, is that when inflammation reaches a clinically significant threshold, it increases the permeability of

the BBB. This increased permeability may disturb the blood-brain metabolic transfer and allow neurotoxic compounds and pathogens, such as viruses [32], to enter the brain. The presence of the unwanted metabolic waste products and other neurotoxic compounds can cause damage to the brain resulting in worsening cognitive performance in certain domains, then cognitive decline and finally dementia. In this model the presence of CRP is necessary, but not sufficient, to cause dementia. This and the fact that serum CRP levels change [33], may explain the observed weak associations between CRP and dementia.

In conclusion, we show in this study is that a single measurement of inflammatory biomarker is associated with worse cognitive performance and future dementia diagnosis, albeit the effects sizes are small. Our results highlight that systemic inflammation might contribute to dementia especially to the vascular type, but with a small effect compared to other well-known risks. Therefore, when evaluating potential mechanistic pathways other factors, such *APOE* genotype and dementia aetiology should also be considered. We also show that although CRP and WBC both measure inflammation, WBC appears to be the better predictor of the two, therefore it may warrant for the use of more complex systemic inflammatory biomarker scores. Nevertheless, although inflammatory biomarkers are not specific for dementia and appear to have a small effect, they may be useful in identifying individuals at risk for inflammation-dysregulation diseases (such as frailty and dementia) possibly at early stage of the condition and might enhance the precision of dementia predicting models.

The strength of our study is the large sample size but is suffers from some limitations.

A potential weakness of this study is that the UKB dataset is not representative of the UK population. Participants tend to live in less socioeconomically deprived areas and adopt better health behaviours (in terms of smoking and drinking) and reported fewer health-related problems, than the population in general [34].

Dementia was identified using hospital episode records (including ICD10 and 9) however we obtained an additional dementia incident cases (38%) from primary care record. The primary care record is only available in 45% of UK Biobank study participants. Therefore, there could be underestimation of true dementia figure in our participant, results presented here could potentially be affected.

In terms of dementia, only 1.12% of the UKB cohort aged 65 and over are identified as having a diagnosis of dementia, which is far lower than the national figure prevalence of dementia—7.1% for the total age 65 and over population (based on 2013 data) [35].

Further limitation of this study is the potential inaccuracy of the dementia diagnoses lacking neuropathologic confirmation.

Finally, the CRP measure was only available for the full sample at baseline, and for a small subsample at follow-up, therefore we cannot comment on its trajectories.

The weak associations found in this healthy volunteer selection bias-prone sample indicates that these effect sizes might be larger considering the general population. Our work also demonstrates that a focus on inflammation might offer opportunities both for the management of future dementia risk and might contribute to the development of early predictor biomarkers for dementia.

## Supporting information

**S1 Table. Associations between inflammatory biomarker score quartiles and baseline cognitive tasks.**
(DOCX)

**S2 Table. Associations between inflammatory biomarker score quartiles and baseline cognitive tasks with the *APOE ε4* non-carrier and carrier subsamples analysed separately.**
(DOCX)

**S3 Table. Associations between inflammatory biomarker score quartiles separately for CRP and WBC and baseline cognitive tasks.**
(DOCX)

**S4 Table. Associations between inflammatory biomarker score quartiles and baseline cognitive tasks additionally adjusted for age squared.**
(DOCX)

**S5 Table. Associations between inflammatory biomarker score quartiles and Instance 2 cognitive tasks.**
(DOCX)

**S6 Table. Cox regression results for inflammatory biomarker score quartiles and dementia risk with the *APOE ε4* non-carrier and carrier subsamples analysed separately.**
(DOCX)

**S7 Table. Cox regression results for separate inflammatory biomarker score quartiles for CRP and WBC and dementia risk.**
(DOCX)

## Acknowledgments

We would like to thank all the UKB participants and staff for making this study possible (Application number 5864).

## Author Contributions

**Conceptualization:** Krisztina Mekli.

**Formal analysis:** Krisztina Mekli, Artitaya Lophatananon, Asri Maharani.

**Funding acquisition:** Kenneth R. Muir.

**Methodology:** Krisztina Mekli, Artitaya Lophatananon, Asri Maharani.

**Supervision:** James Y. Nazroo, Kenneth R. Muir.

**Writing – original draft:** Krisztina Mekli.

**Writing – review & editing:** James Y. Nazroo.

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
