## [Decision Letter · Decision Letter 0]

10 Jan 2023

PONE-D-22-33736Association between an inflammatory biomarker score and future dementia diagnosis in the population-based UK Biobank cohort of 500,000 peoplePLOS ONE

Dear Dr. Mekli,

Thank you for submitting your manuscript to PLOS ONE. After careful consideration, we feel that it has merit but does not fully meet PLOS ONE’s publication criteria as it currently stands. Therefore, we invite you to submit a revised version of the manuscript that addresses the points raised during the review process.

Based on the reviewers' suggestions, the paper needs major revision.  The reviewers' comments can be found below.

We look forward to receiving your revised manuscript.

Kind regards,

Tanja Grubić Kezele, Ph.D., M.D.

Academic Editor

PLOS ONE

Reviewers' comments:

Reviewer's Responses to Questions

**Comments to the Author**

1. Is the manuscript technically sound, and do the data support the conclusions?

Reviewer #1: Yes

Reviewer #2: Yes

Reviewer #3: Yes

2. Has the statistical analysis been performed appropriately and rigorously? 

Reviewer #1: No

Reviewer #2: Yes

Reviewer #3: No

3. Have the authors made all data underlying the findings in their manuscript fully available?

Reviewer #1: Yes

Reviewer #2: Yes

Reviewer #3: Yes

4. Is the manuscript presented in an intelligible fashion and written in standard English?

Reviewer #1: Yes

Reviewer #2: Yes

Reviewer #3: Yes

5. Review Comments to the Author

Reviewer #1: The authors used data from the UK Biobank to examine the association of a composite of inflammatory biomarkers measured in blood with concurrent cognition (measured across 5 cognitive domains) and incident dementia risk over an 11-year follow-up period. In this sample of 474,233 participants, the authors found that elevated inflammatory biomarker level was associated with worse performance on measures of prospective memory, fluid intelligence, and reaction time. This is a great data set and the authors set out to answer an important question. With the large amount of information available through UKB, there are a number of steps the authors could take to improve their study design and the interpretability of their findings. I provide specific critiques below.

In the intro the authors discuss the inconsistent findings from studies looking at immune markers in AD and MCI individuals and state that these inconsistencies make it difficult to develop validated biomarkers of early diagnosis and disease. Two points that I think the authors should address in conjunction with this statement. 1) there have been several fairly recent meta-analyses on this topic which show fairly consistent associations for a number of inflammatory or anti-inflammatory proteins. These meta-analyses have nearly 200 studies in them. Simply saying the results are inconsistent without providing some summary of the many studies on this topic doesn’t help advance the readers understanding of a well established area of research. 2) immune biomarkers will likely never be used as true biomarkers for AD. They’re not specific to any one neurodegenerative etiology and will unlikely be able to outperform well validated measures such as amyloid-beta 42 and soluble ptau217 that are disease-specific and represent indicators of AD’s defining features. That said, I do however think it’s immensely important to study inflammatory proteins in blood and in CSF to understand more about the underlying disease biology and he role of immunity.

Inflammatory proteins have also been implicated – perhaps even more strongly – in vascular cognitive impairment and vascular dementia. This should be acknowledged in the introduction, especially that 1) the authors are examining all-cause dementia as an outcome, and 2) because of the high rates of cerebrovascular disease in patients with Alzheimer’s disease.

Some rationale for the reason to exclude participants with APOEe2/APOEe4 should be provided in the genetics data section.

Age, sex, and APOEe4 were the only covariates included in the author’s regression models. However, there are many other potential confounders known to both influence inflammation and cognition/dementia that should be considered. Race, education (or another SES proxy), as well as cardiovascular risk factors should be considered – perhaps in a different model – as additional covariates. Without this sort of adjustment, the findings and challenging to interpret.

Related to the comment above, Table 1 should be expanded to provide the reader with more information about the study sample. Things like race, education, APOEe4 status, clinical diagnoses (hypertension, diabetes) should be included. I also suggest splitting Table 1 by inflammatory biomarker quartile rather than by control vs. dementia, which is less informative.

Was time-to-event data available to look at dementia risk using Cox models. I imagine this would be a more powerful way of detecting effects. The fact that people with dementia within 3 years of the baseline visit were excluded suggests that the date of dementia onset can be determined.

The authors are encouraged to look at dementia risk by etiology. The relationship between inflammatory composite score and Alzheimer’s dementia and vascular dementia should be examined.

Because there are only 2 proteins in the composite score, it makes sense to look at them individually as well to see if either has an especially strong or especially weak association with the outcomes.

The authors have APOEe4 information. It would be informative to know whether the biomarker-cognition and biomarker-dementia relationships differed by APOEe4 possession status.

The authors should provide a rationale for adjusting for baseline cognitive variables in their examination of proteins in relation to dementia risk.

Reviewer #2: This research article titled “Association between an inflammatory biomarker score and future dementia diagnosis in the

population-based UK Biobank cohort of 500,000 people” by Mekli et al. Peripheral inflammation is an important topic in the context of dementia prevention and has shown to be related to cognitive impairments. This research study investigated peripheral inflammation for cognitive impairment and dementia risk. Although it is somewhat confirmatory study, it provided important information from a huge real world cohort. There are only some comments from me:

1. The major limitation for the study was cross-sectional for cognitive impairment.

2. There was not enough discussion on why peripheral inflammation was associated with better performance of pair match. This was a cross-sectional finding, and it is unknown if they are still positively associated in a follow-up.

3. The time between blood tests and cognitive tests were variable for different participants and should be controlled in the analyses.

Reviewer #3: The authors assessed the association between a composite inflammatory biomarker score and five cognitive scores and also the development of dementia in following 3-11 years using data from the UK biobank.

I have some questions about the data analysis.

In the regression models, was age entered as a continuous variable? The relationship with age may not be linear, a spline or other functions of age may be needed in the model. What about the interaction between age and sex? Is it associated with the outcomes?

There seems to be a trend in the effects of the quartiles. What if the composite biomarker score is used as a continuous variable instead of quartiles?

A cox model is a more appropriate model for the incident dementia risk analysis instead of a logistic model. Age can be used as the primary time scale.

How correlated are the five cognitive scores? It might help us to understand why the pair matching result seemed counterintuitive.

Have the authors analyzed the two components of the composite biomarker score separately to see if one may contribute to the association more?

Individuals outside 3 SD of the scores were not included in the analysis. What proportion of the samples were in this outlier group?

Table 1 should include descriptive statistics of more variables, such as APOE genotype, the five cognitive scores. Sample size should be included as well. Better to show row percentages instead of column percentages.

6. PLOS authors have the option to publish the peer review history of their article (what does this mean?). If published, this will include your full peer review and any attached files.

Reviewer #1: No

Reviewer #2: No

Reviewer #3: No

---

## [Author Response · Author response to Decision Letter 0]

14 Apr 2023

We thank the Reviewers for their comments and recommendations to significantly improve the quality of our manuscript. We would like to draw their attention to the Response to Reviewers document, in which address each of the suggestions.

---

## [Decision Letter · Decision Letter 1]

2 May 2023

PONE-D-22-33736R1Association between an inflammatory biomarker score and future dementia diagnosis in the population-based UK Biobank cohort of 500,000 peoplePLOS ONE

Dear Dr. Mekli,

Thank you for submitting your manuscript to PLOS ONE. After careful consideration, we feel that it has merit but does not fully meet PLOS ONE’s publication criteria as it currently stands. Therefore, we invite you to submit a revised version of the manuscript that addresses the points raised during the review process.

Your manuscript, entitled "Association between an inflammatory biomarker score and future dementia diagnosis in the population-based UK Biobank cohort of 500,000 people", has been reviewed. Your efforts to revise the manuscript are appreciated. However, the peer review process continues because Reviewer 3 has a few additional comments that the author should address. Please find the reviewer's commentary below.

We look forward to receiving your revised manuscript.

Kind regards,

Tanja Grubić Kezele, Ph.D., M.D.

Academic Editor

PLOS ONE

Reviewers' comments:

Reviewer's Responses to Questions

**Comments to the Author**

1. If the authors have adequately addressed your comments raised in a previous round of review and you feel that this manuscript is now acceptable for publication, you may indicate that here to bypass the “Comments to the Author” section, enter your conflict of interest statement in the “Confidential to Editor” section, and submit your "Accept" recommendation.

Reviewer #1: All comments have been addressed

Reviewer #2: All comments have been addressed

Reviewer #3: (No Response)

2. Is the manuscript technically sound, and do the data support the conclusions?

Reviewer #1: Yes

Reviewer #2: Yes

Reviewer #3: Yes

3. Has the statistical analysis been performed appropriately and rigorously? 

Reviewer #1: Yes

Reviewer #2: Yes

Reviewer #3: Yes

4. Have the authors made all data underlying the findings in their manuscript fully available?

Reviewer #1: Yes

Reviewer #2: Yes

Reviewer #3: Yes

5. Is the manuscript presented in an intelligible fashion and written in standard English?

Reviewer #1: Yes

Reviewer #2: Yes

Reviewer #3: Yes

6. Review Comments to the Author

Reviewer #1: Thank you for your responsive revision. I have no further comments.

Reviewer #2: (No Response)

Reviewer #3: I have a few additional comments for this revision.

Estimates from models stratified by APOE ε4 carrier status were reported. However, stratified analysis is not necessary if the effects are not different between APOE ε4 carriers vs non-carriers. A formal testing of the quartiles by carrier status can be performed and only results based on the combined data need to be shown if there are no interaction effects.

A case-only approach can be used to evaluate if effects of inflammatory biomarker score were different for vascular dementia vs AD (ref: Wang et al Stat Med. 2016 Feb 28; 35(5): 782–800). The model will only include dementia cases (from both subtypes) and the response variable is a binary indicator variable for AD (or vascular dementia) status. A logistic model can be used to see if the inflammatory biomarker score is associated with dementia subtype while adjusting for other covariates. Under the null of no difference between the two subtypes there should be no association.

This sentence: “The respondents were censored at the year dementia was first recorded or death, whichever came first.”- dementia is an event not a censoring.

7. PLOS authors have the option to publish the peer review history of their article (what does this mean?). If published, this will include your full peer review and any attached files.

Reviewer #1: No

Reviewer #2: No

Reviewer #3: No

---

## [Author Response · Author response to Decision Letter 1]

5 Jun 2023

I would like to thank the Reviewers for reading the revised manuscript and providing us with further comments and recommendations. Please find our responses in the attached Response to Reviewers document. 

Sincerely, Krisztina Mekli

---

## [Decision Letter · Decision Letter 2]

9 Jun 2023

PONE-D-22-33736R2Association between an inflammatory biomarker score and future dementia diagnosis in the population-based UK Biobank cohort of 500,000 peoplePLOS ONE

Dear Dr. Mekli,

Thank you for submitting your manuscript to PLOS ONE. After careful consideration, we feel that it has merit but does not fully meet PLOS ONE’s publication criteria as it currently stands. Therefore, we invite you to submit a revised version of the manuscript that addresses the points raised during the review process.

Your manuscript, entitled "Association between an inflammatory biomarker score and future dementia diagnosis in the population-based UK Biobank cohort of 500,000 people", has been reviewed. Your efforts to revise the manuscript are appreciated. However, the peer review process continues because Reviewer 3 has one additional comment that the author should address. Please find the reviewer's commentary below.

We look forward to receiving your revised manuscript.

Kind regards,

Tanja Grubić Kezele, Ph.D., M.D.

Academic Editor

PLOS ONE

Journal Requirements:

Reviewers' comments:

Reviewer's Responses to Questions

**Comments to the Author**

1. If the authors have adequately addressed your comments raised in a previous round of review and you feel that this manuscript is now acceptable for publication, you may indicate that here to bypass the “Comments to the Author” section, enter your conflict of interest statement in the “Confidential to Editor” section, and submit your "Accept" recommendation.

Reviewer #2: All comments have been addressed

Reviewer #3: All comments have been addressed

2. Is the manuscript technically sound, and do the data support the conclusions?

Reviewer #2: Yes

Reviewer #3: Yes

3. Has the statistical analysis been performed appropriately and rigorously? 

Reviewer #2: Yes

Reviewer #3: Yes

4. Have the authors made all data underlying the findings in their manuscript fully available?

Reviewer #2: Yes

Reviewer #3: Yes

5. Is the manuscript presented in an intelligible fashion and written in standard English?

Reviewer #2: Yes

Reviewer #3: Yes

6. Review Comments to the Author

Reviewer #2: (No Response)

Reviewer #3: This revision has addressed all previous comments. I have one additional comment- the following sentence in the abstract is not quite accurate: “This association was probably driven by the vascular dementia cases in the samples (HR=1.483, p=0.055)”. The case-only approach is to assess if there is disease subtype heterogeneity. The marginally significant p value suggests that the association with highest biomarker quartile probably varies between disease subtypes (vascular dementia and Alzheimer’s disease).

7. PLOS authors have the option to publish the peer review history of their article (what does this mean?). If published, this will include your full peer review and any attached files.

Reviewer #2: No

Reviewer #3: No

---

## [Author Response · Author response to Decision Letter 2]

15 Jun 2023

Please see our responses in the attached Response to Reviewers document. Thank you very much.

---

## [Decision Letter · Decision Letter 3]

19 Jun 2023

Association between an inflammatory biomarker score and future dementia diagnosis in the population-based UK Biobank cohort of 500,000 people

PONE-D-22-33736R3

Dear Dr. Mekli,

We’re pleased to inform you that your manuscript has been judged scientifically suitable for publication and will be formally accepted for publication once it meets all outstanding technical requirements.

Kind regards,

Tanja Grubić Kezele, Ph.D., M.D.

Academic Editor

PLOS ONE

Additional Editor Comments (optional):

Reviewers' comments:

Reviewer's Responses to Questions

**Comments to the Author**

1. If the authors have adequately addressed your comments raised in a previous round of review and you feel that this manuscript is now acceptable for publication, you may indicate that here to bypass the “Comments to the Author” section, enter your conflict of interest statement in the “Confidential to Editor” section, and submit your "Accept" recommendation.

Reviewer #3: All comments have been addressed

Reviewer #4: All comments have been addressed

2. Is the manuscript technically sound, and do the data support the conclusions?

Reviewer #3: (No Response)

Reviewer #4: Yes

3. Has the statistical analysis been performed appropriately and rigorously? 

Reviewer #3: (No Response)

Reviewer #4: Yes

4. Have the authors made all data underlying the findings in their manuscript fully available?

Reviewer #3: (No Response)

Reviewer #4: Yes

5. Is the manuscript presented in an intelligible fashion and written in standard English?

Reviewer #3: (No Response)

Reviewer #4: Yes

6. Review Comments to the Author

Reviewer #3: (No Response)

Reviewer #4: The authors have addressed all necessary comments. All parts of the manuscript are acceptable for publication.

7. PLOS authors have the option to publish the peer review history of their article (what does this mean?). If published, this will include your full peer review and any attached files.

Reviewer #3: No

Reviewer #4: No

---

## [Editor Report · Acceptance letter]

26 Jun 2023

PONE-D-22-33736R3 

Association between an inflammatory biomarker score and future dementia diagnosis in the population-based UK Biobank cohort of 500,000 people 

Dear Dr. Mekli:

I'm pleased to inform you that your manuscript has been deemed suitable for publication in PLOS ONE. Congratulations! Your manuscript is now with our production department. 

Kind regards, 

on behalf of

Prof. dr. Tanja Grubić Kezele 

Academic Editor

PLOS ONE